# Epidemiology of Acute Injuries in Surfing: Type, Location, Mechanism, Severity, and Incidence: A Systematic Review

**DOI:** 10.3390/sports8020025

**Published:** 2020-02-20

**Authors:** Katherine McArthur, Darcy Jorgensen, Mike Climstein, James Furness

**Affiliations:** 1Water Based Research Unit, Faculty of Health Sciences, Bond University, Gold Coast 4207, Australia; katherine.mcarthur@student.bond.edu.au (K.M.); darcy.jorgensen@student.bond.edu.au (D.J.); michael.climstein@scu.edu.au (M.C.); 2Clinical Exercise Physiology, School of Health and Human Sciences, Southern Cross University, Bilinga 56155, Australia; 3Physical Activity, Lifestyle, Ageing and Wellbeing Research Group, Faculty of Health Sciences, University of Sydney, Lidcombe 2141, Australia

**Keywords:** surfing, injury, epidemiology, acute

## Abstract

Prospective and retrospective studies have examined traumatic injuries within competitive and recreational surfers worldwide using online surveys and health care facility (HCF; e.g., hospital, emergency department, medical record) data. However, few studies have provided a synthesis of all available literature. The purpose of this study was to obtain, critique and synthesise all literature specific to acute surfing injuries, and evaluate differences in injury type, mechanism and location between HCF and survey data. A systematic literature review design was used to identify relevant articles from three major databases. Peer-reviewed epidemiological studies of musculoskeletal surfing injuries were included. A modified AXIS tool was used for critical appraisal, and objective data was extracted and synthesized by lead researchers. Overall frequencies for injury location, type and mechanism were calculated from raw injury data. A total of 19 cross-sectional articles of fair to good quality (Modified AXIS 54.2–83.3%) were included in this study; 17 were National Health and Medical Research Council (NHMRC) level III-2 (retrospective) and two were level II (prospective). Articles examined competitive, recreational and combined populations. Injury data from Australia, Brazil, UK, USA, Portugal, Japan, Norway, and worldwide were represented. Skin (46.0%; HCF 50.1%, survey 43.8%) and being struck by own surfboard (38.6%; HCF 73.4%, survey 36.7%) were the most common injury type and mechanism. Head, face and neck injuries were most common in HCF (43.1%) versus lower limb injuries (36.4%) in survey data. Incidence proportion was highest in aerialists (0.48). Incidence rate (number of injuries per 1000 h) ranged from 0.74 in Australian surfers (Melbourne) to 6.6 in international contest surfers from medical record data. This review highlights the prevalence of skin, board-related, head, face and neck, and lower limb surfing injuries across available literature. Proposed use of protective equipment and foam-based surfboards in dangerous or crowded surf locations may reduce injury risk.

## 1. Introduction

Surfing, from its origins in ancient Polynesia, has grown over the years to become part of many cultures around the world [1]. Recreational and competitive surfing has increased in popularity to an estimated 37 million surfers worldwide and an estimated 2.7 million surfers here in Australia [2,3]. Surfing is an international sport and will be featured in the Olympic Games for the first time in 2020 [4]. The growing popularity and accessibility of wave pools such as the World Surf League (WSL) wave pool, illustrates the potential for surfing to grow exponentially in non-coastal locations. The sport of surfing has diversified in surfing environment, equipment and style, including the introduction of mechanical wave pools, big wave surfing, and aerial manoeuvres which all present their unique thrills and hazards to those participating.

Injury is a risk with participation in all sports (recreationally and competitively), and surfing is no exception [5,6]. Uncontrolled and often unpredictable ocean environments present unique hazards to surfers; sand, coral reef and rock breaks, water depth, wave size and type, water temperature, presence of other surfers, and local marine animals all contribute to injury risk [5,7,8,9]. Surfing equipment such as wetsuits, booties and gloves, although designed for thermoregulation and maintaining body heat, have indirectly also helped protect surfers from lacerations and contusions from rock and reef [9]. The invention of surfboard leashes has also helped protect surfers from being hit by others’ surfboards during wipeouts however, presents new hazards to board riders from elastic recoil [6]. Advances in technology have made surfboards lighter and more maneuverable with changes to board shape and use of fins, allowing quicker changes in direction and aerial maneuvers to be incorporated into surf style [5,6,7,10]. To reduce injury occurrence and continue the progression of the sport, a thorough understanding of injury location, types and mechanisms is needed [11].

To date, many studies have examined acute surfing injuries in recreational and competitive surfers both prospectively and retrospectively across the globe. Information on injury type, location, severity and mechanism has been documented from the earliest study by Allen et al. [12] who examined acute surfing injuries at Waikiki, Hawaii to more recent studies by Burgess et al. [13], Dimmick et al. [6], Hohn et al. [14] and Inada et al. [15] examining acute surfing injuries in Australia (New South Wales), Australia (Queensland), globally from WSL data and in Japan respectively. Various studies have provided data from a number of sources; online surveys, emergency departments (ED) and medical records have been used to collect primarily retrospective data [5,6,7,8,9,10,12,13,14,15,16,17,18,19,20,21,22,23,24,25,26]. Medical services at surfing competitions have also been used to collect prospective data, however, to our knowledge only two studies to date have been conducted so far [6,27]. While all of these research methods profile surfing injuries, online surveys are limited to recall bias, and EDs or hospitals only represent injuries serious enough where the participant seeks emergency medical attention [20].

To our knowledge only two studies have attempted to collate and synthesise some of the acute surfing injury data, Donosa and Cabral [28], and Nathanson [29]. While both studies summarised injury type, location and mechanism data from six and 11 studies respectively, several epidemiological studies were not included [6,7,12,13,14,15,18,20,24,26] Furthermore, there have been numerous epidemiological studies published following the 2015 publication by Donosa [6,7,13,14,15,18,20,24,26]. Therefore, an updated review of the literature is needed to gain a comprehensive understanding of acute surfing injuries around the world, between various populations (ages and competition levels), and through different data collection settings.

This paper aimed to obtain, critique and synthesise all the available literature specific to acute surfing injuries, and evaluate the differences in injury type, mechanism and location between survey and health care facility (HCF) data. This paper will be the first large scale review of acute surfing injury literature and the first comparison of HCF and survey reported injuries in surfing. This information gained from our findings could be used to develop injury prevention strategies for the growing surfing population.

## 2. Materials and Methods

The methodological design of this systematic literature review was in alignment with the Preferred Reporting Items for Systematic Reviews and Meta-Analyses (PRISMA) statement to ensure transparency and appropriate reporting [22].

### 2.1. Literature Search

For this critical review, a systematic search of key databases was completed in September 2018. To identify appropriate search terms to best depict our research proposal, a preliminary rapid search of the available literature was performed using the key words ‘surfing’ and ‘injury’ in PubMed. Once a comprehensive understanding of the available literature was established, key terms of the research question were converted into a search strategy. Expansions of the terms using Boolean operators (surf OR surfing) AND (acute OR trauma) AND (injury OR injuries) were used to amass literature. All studies, regardless of publication date, that met the inclusion criteria were included. 

The PubMed full search strategy was as follows: ((surfers OR surfer OR surfing OR surf OR surfboard OR surfboarding OR “surfboard riding” OR “water sport” OR “water sports” OR “Water Sports”[Mesh]) AND (injury OR injuries OR “Wounds and Injuries”[Mesh] OR trauma OR traumatic OR “first aid” OR accident OR accidents OR disability OR disabilities OR “Cumulative Trauma Disorders”[Mesh] OR emergency OR fracture OR fractures OR “Fractures, Avulsion”[Mesh])).

Using the proposed strategy, the PubMed, Embase, CINAHL and SPORTDiscus databases were compiled by two independent authors into two independent citation databases (EndNote X8, Thompson Reuters, New York, NY, USA).

### 2.2. Study Selection

Two authors independently selected and screened all imported studies to limits search bias, duplication bias, inclusion bias, and selector bias [30]. Following the removal of duplicates, articles were screened for eligibility by title and abstract, full details of study selection criteria are provided in Table 1. The inclusion and exclusion criteria were established prior to database searches. Full-texts were required for potentially eligible studies or if a paucity of information could be retrieved from title and abstract only. The inclusion and exclusion criteria (Table 1) were again stringently applied to determine final eligibility. 

For ensuring the transparent and complete reporting of study selection, PRISMA statement for reporting systematic reviews and meta-analyses [22] were used, and an illustration of search results is described by the PRISMA flow diagram (Figure 1). 

To be eligible, injuries must have been definable as acute in their stage: an acute injury has been defined as a sudden onset of sharp pain or sudden impact that the person can relate to a specific situation, normally resulting in tissue damage in a localized region, with persistent or episodic pain lasting less than three months [31]. Studies were excluded if the injury could not be defined as acute or if the injury was non-musculoskeletal in nature. Case series were excluded from this study as they do not depict a true epidemiological representation of all surfing related injury incidence, location, types and mechanisms.

### 2.3. Critical Appraisal

Critical appraisal systematically assessed research papers judge the reliability of the study being presented in the paper. The Appraisal Tool for Cross-Sectional Studies (AXIS) was chosen as it was developed for use in appraising observational cross-sectional studies [32]. Using a modified version of the Appraisal Tool for Cross-Sectional Studies (AXIS), two authors independently assessed the quality of the included articles. The modification of an appraisal tool has been previously established and shown to be effective [21,33]. Researchers added five questions (12–16, inclusive) to the original AXIS Critical Appraisal Tool to identify and appraise whether certain epidemiological data was included in the studies (see Appendix A for Modified AXIS tool). Question 12 permitted a ‘2′ point answer if the study classified injury using body region, type, and stage of injury, whereas a score of ‘1′ if the injury was classified using at least one of body region, type, and/or stage of injury, or a score of ‘0′ if no injury classification was used. Question 15 permitted a ‘2′ point score if injury data was collected prospectively, opposed to a ‘1’ point score if injury data was recalled in retrospect. Self-reported retrospective data is vulnerable to recall bias, which represents a major threat to the internal validity and credibility of a study [34]. Other questions regarding non-responders was deemed not applicable when appraising studies collecting data retrospectively from ED and hospital records and was removed from total Modified AXIS score. A score of ‘1’ was assigned to a ‘yes’ answer and ‘0’ for a ‘no’ answer. If a question was deemed ‘Non Applicable’, it was discredited from the overall score, thus not affecting the articles overall appraisal score. The maximum attainable raw score of this modified AXIS was 27. The modified AXIS used a checklist of twenty-five questions opposed to the original twenty. The quality of individual articles was assessed based on the Downs and Black Checklist scoring system; whereby the Downs and Black raw scores were converted into percentages for the scoring system and modified to be used with our Modified AXIS tool [18]. A score equal to or greater than 74% was considered ‘good’ quality, a score between 55%–73.9% was considered ‘fair’ quality, and a score less than 54.9% was considered ‘poor’ quality articles [35]. 

Two authors (K.M. and D.J.) individually appraised eligible articles. To determine the interrater reliability of the two author’s appraisal scores, a Cohen’s Kappa Coefficient (k) was derived through SPSS statistical package (IBM SPSS Statistics 25.0, International Business Machines Corporation, New York, United States). A final appraisal was completed by the two authors (K.M., D.J.) to resolve any discrepancies, where a final appraisal score was decided upon. To determine the methodological quality of the studies, the Kennelly rating system for critical appraisals was applied [35].

### 2.4. Data Extraction 

Following the collection of eligible studies, relevant data was extracted and tabulated. Such data included: author, title, aims, research design, level of the evidence, participants, study setting, type of injury, severity of injury, mechanism of injury, incidence proportion (IP), incidence rate (IR), and the studies main findings. The level of the evidence was established by The National Health and Medical Research Council (NHMRC) levels of evidence, and critical appraisal scores described by the modified-AXIS tool and corresponding Kennelly rating [35,36].

As described by Furness et al. [18], risk and rates of injury are two definitions to quantitatively measure injury incidence. Incidence proportion (IP) was defined as the probability of an athlete getting injured over a 12-month period. Incidence rate (IR) was defined as the incidence of injury over a set unit of exposure, often 1000 h of surfing or 1000 surf days. Where a study used a retrospective cohort survey design, that had a specified time frame to which the injury occurred i.e., in the last 12-months, and when surfing time had been quantified, IR was be calculated if not already provided. If a study reported IR in 1000 surfing days, this was converted to 1000 h by using the mean duration of surf time multiplied by 1000. In the instances where IP was not provided this was calculated providing the total number of athletes exposed to the risk was known.

### 2.5. Data Synthesis/Analysis

To report on mass frequency data, the most appropriate method determined by the research team was to collate the absolute values for injury types, body regions injured and mechanisms of injury (i.e., frequency data for one specific injury type at one location was not provided). This data was presented for each study and provided in a tabular format. In addition to this, all absolute frequency values for injury location were summed together and divided by the total number of injuries sustained to provide an overall frequency value for each body location. This was illustrated in a graphical format and permitted an overall frequency for all homogenous data. Special attention was placed on distinguishing the differences between body regions commonly affected and the research setting, such as a hospital emergency department or online survey. The assumption being that more acute lacerations, skin injuries and/or being struck by own board injuries would be captured by health facility data versus survey data showing higher soft tissue and/or manoeuvre-based injuries that would most likely present to other outpatient facilities (general practitioner, physiotherapy clinics, etc.). Additionally, injury analysis was attempted between recreational and professional surfers, short board and long board riders, as well as geographical surf locations.

## 3. Results

### 3.1. Data Search Results

The PRIMSA flow diagram (Figure 1) illustrates the total number of studies compiled through database searching and expert opinion, study removal for duplication, and screening processes. A total of 6513 studies were exported to the respective reference management system libraries. Following the removal of duplicates (269 duplicates removed), 6244 articles were screened by title and abstract for eligibility, where a further 6213 were excluded. Full-text was required for 31 studies, where a further 12 were excluded with reasons. Reasons included the following: four studies documented discussions of injury trends or reviews of surfing injury epidemiology literature, three studies did not relate to the sport of surfboard riding (e.g., used “surfing” as a play-on-words in the title) and five studies were excluded as the surf injury epidemiology data could not be separated from other surf craft or swimming injury data. A total 19 studies were therefore included for final data extraction, analysis and discussion.

### 3.2. Critical Appraisal Results

Of the 19 articles included, 10 were rated ‘fair’ quality and nine were rated ‘good’ quality, mean quality rating was 71.5% (SD ± 11%) and range between 54.2% and 95.8% (Table 2).

No articles were rated poor quality. The Cohen’s Kappa yielded a ‘moderate agreement’ between raters on initial critical appraisal, raters resolved any differenced by discussion to agree on a score for final critical appraisal of all 19 articles (initial k = 0.566; final k = 1.000). The critical appraisal results are shown in Table 2 as raw scores and percentages along with their quality rating.

The AXIS critical appraisal tool provides no scoring system differentiating between poor, fair or good quality articles as the Downs and Black does and was identified as a limitation to the use of this tool [32]. Other authors conducting systematic reviews have modified the Downs and Black checklist to add additional questions and subsequently modified the quality scoring system thus similar adaptations were used for the purpose of this review [37].

### 3.3. Geographical Location of Injury Research

The literature on acute surfing injuries outlines injury data from many countries around the world and presents both local and global data. Of the studies included in this literature review, acute injury data from eight countries was represented among 16 studies and three studies collected data globally [14,25,27]. Of the 16 studies from specific countries, six were from Australia, three were from Brazil, two were from the UK, two were from the USA, and Japan, Portugal and Norway each had one study (Figure 2). Global study distribution is further represented in Table 3.

### 3.4. Study Characteristics and Main Findings

The majority of studies were cross-sectional retrospective cohort studies using online surveys, questionnaires or medical records to collect data. Only two of the included studies were cross-sectional prospective cohort studies, using medical record data and a hospital emergency department (ED) questionnaire for data collection. The publication years ranged from 1977 to 2018, with the majority taking place after the year 2010. Further study characteristics are summarized in Table 3. The language used to describe injury type, location and mechanism was largely consistent amongst the literature. Descriptors for injury type were the most diverse across the literature, indicating a need for grouping for simplicity in this review. For the purpose of this review, categories within injury type were grouped in such a way that findings could be compared across the available literature; “skin” injuries are inclusive of abrasions, lacerations, burns, haematoma and contusions, “soft tissue” injuries are inclusive of muscular strain, muscle cramping, ligament sprain/rupture, tendon sprain/rupture and tendonitis, and “joint” injuries are inclusive of dislocation, subluxation, cartilage disruption, meniscus tear, bursitis and vertebrae/facet injuries. These broad injury type groupings were based upon previous epidemiological study designs [10,18].

### 3.5. Injury Incidence Rate and Injury Proportion

Injury incidence proportions were poorly represented in the included literature in only 2 studies. Incidence proportions were reported individually for aerialist (0.48), competitive (0.42) and recreational (0.35) surfers in Furness et al. [18], whereas Ulkestad et al. [9] reported an incidence proportion of 2.1 for a combined population. It is worth noting Ulkestad et al. [9] collected data from surfers in Norway and Furness et al. [18] collected data from surfers largely in Australia which could account for the difference in injury incidence proportion given the environmental difference in climate; cold weather exposes surfers to risk of hypothermia but also requires the use of wetsuits which provide sun protection, aid flotation and protect against abrasions. Injury incidence rates and proportions per study are included in Figure 3. Donosa and Cabral [28] did not discuss incidence proportion in their study, and Nathanson [29] did not discuss surfing injury incidence rate or incidence proportion.

### 3.6. Type of Injury

This review found the most common type of injury was to the skin representing 46% of total injuries, followed by soft tissue injuries (22.6%) and bone injuries (9.6%). This was comparable in health care facility and survey data showing skin injuries representing 50.1% and 43.8% respectively, also followed by soft tissue injuries (HCF 17.6%, survey 25.4%) and bone injuries (HCF 11.9%, survey 8.4%). Skin injury percentages in individual studies were largely similar to our combined data with 12 of the included studies skin injury percentages ranging between 44% and 56.2% [2,5,22]. Overall variation in skin injury percentages between studies was from 17.2% in Furness et al. [18] to 75% in De Moraes et al. [17]. Hohn et al. [14] was the only study which did not include a category for skin injuries. Full breakdown of types of injuries represented in the literature can be seen in Figure 4. In Figure 4, skin injuries include abrasion, laceration, burn, haematoma, and contusions; soft tissue injuries include muscular strain, muscular cramping, ligament sprain/rupture, tendon sprain/rupture, and tendonitis; joint injuries include dislocations, subluxation, cartilage disruption, meniscus tear, bursitis, and vertebrae facet injury. The following injury types represented less than 1.0% of total injuries and were therefore not include in Figure 4: peripheral nerve injuries represented 0.9% of total injuries (HCF 0%, survey 1.3%), ear perforation 0.7% (HCF 0%, survey 1.0%), marine life injuries 0.3% (HCF 0%, survey 0.5%), hypothermia 0.2% (HCF 0%, survey 0.3%), dental injuries 0.1% (HCF 0%, survey 0.1%), and spinal cord injuries 0% (HCF 0%, survey 0%).

### 3.7. Location of Injury

The most common body region injured was the face, head and neck (33.8%) followed closely by lower limb (33.0%). Face, head and neck represented 43.1% of injuries in HCF studies and 27.9% of survey studies, whereas lower limb injuries represented 27.8% of HCF studies and 36.4% of survey studies. Upper limb/arm injuries were very similarly reported between both HCF and survey studies at 16.5% and 16.8% respectively. Full breakdown of injury locations represented in the literature can be seen in Figure 5.

### 3.8. Mechanism of Injury

This review found surfers being struck by their own board to be the most common cause of injury (38.5%) followed by approaching a wave or performing a manoeuvre while surfing (20.3%) and striking the seafloor or sea surface (18.4%). These results were comparable to ED and survey data where surfers being struck by own board represented 73.4% and 36.7% of injuries respectively. Injury mechanism for both combined, and survey and HCF data are presented in Figure 6. Table 4 summarizes most common injury type, body region affected and mechanism as reported by each individual study included in this review, while Appendix B summarizes further detail on injury type, body region affected and mechanism for each included study. Full breakdown of injury mechanisms represented in the literature can be seen in Figure 6. The following injury mechanisms represented less than 1.0% of total injuries and were therefore not include in Figure 6: walking on beach/rock/reef represented 0.3% of total injuries (HCF 0%, survey 0.3%), low temperature 0.3% (HCF 0%, survey 0.3%), floater/water vessel 0.5% (HCF 0%, 0.5%), and fin/rope 0.9% (HCF 0%, survey 0.9%).

## 4. Discussion

The aim of this systematic review was to obtain, critique and synthesise all the available literature specific to acute surfing injuries, and evaluate the differences in injury type, mechanism and location between survey and health care facility data. With an average methodological rating of 75.1%, the studies selected for this review are of good quality and likely provide an accurate representation of acute surfing injuries sustained within their selected geographic region.

Previous studies by Donosa and Cabral [28], and Nathanson [29] have collated and synthesised acute surfing injury literature. Donosa and Cabral [28] reviewed surfing injury type, body region and mechanism across 11 studies and Nathanson [29] reviewed acute surfing injuries among five outpatient and six hospital-based studies. Our results were comparable to those found by Donosa and Cabral [28] and Nathanson [29]; however, we were able to synthesize data from a greater number of studies and examined differences between prospective and retrospective studies, and differences between data collected from surveys and questionnaires compared to health care facility data.

### 4.1. Injury Incidence Rate and Injury Proportion

Injury incidence rates (IR) were moderately well represented in the literature, reported in nine of the 19 included studies. The variability in incidence rates can be explained by both differences in data collection methods (online survey, questionnaire or health care facility data), prospective compared with retrospective data, and competitive status of surfers. Overall incidence rates (studies grouping both competitive and recreational surfers) were similar, ranging from 0.74 to 1.79 injuries per 1000 h surfed [10,31]. All studies reporting incidence rates were retrospective using data from either online surveys or interview questionnaires. Furness et al. [18] was the only study isolating recreational surfers whose incidence rate was reported 2.18 injuries per 1000 h surfed. Incidence rates in competitive surfers also showed large variability, ranging from 0.3 injuries to 13 injuries per 1000 h surfed [5,27]. Incidence rates collected by Nathanson [27] and Inada [15] both reported 6.6 injuries per 1000 h surfed; both studies collected data from medical records; Nathanson [27] collecting data retrospectively from surfing contests and Inada [15] retrospectively also from surfing contests. Questionnaire, interview and online survey collecting retrospective data from competitive surfers had lower incidence rates of 0.3, 1.08 and 1.51 respectively [5,8,31]. Furness et al. [18] also separated surfers who performed aerial maneuvers from competitive surfers and reported an incidence rate of 1.35 injuries per 1000 h surfed. 

Donosa and Cabral [28], in their review, briefly discuss injury incidence in surfing however, report incidence in number of injuries per 1000 surfing days as opposed to number of surfing injuries per 1000 h surfed as discussed in this review. Donosa and Cabral [28] reported the Lowdon et al. [23] acute injury rate of 3.5 injuries per 1000 surfing days and the Taylor et al. [10] injury rate of 2.2 injuries per 1000 surfing days, compared to the number of injuries per 1000 h surfed reported in this review (see Figure 2). Donosa and Cabral [28] reported surfing as a relatively safe sport with injury incidence rates lower when compared to other sports such as skateboard. With injury incidence rates ranging between 0.3 and 13 as found in this review, data shows variability in incidence rate of surfing injuries and suggests surfing may not be as safe as previously reported [5,27]. 

### 4.2. Type of Injury

There was slight variation in the reporting of “skin”, “soft tissue” and “joint” injuries in the literature. For example, thorough separation of categories such as abrasion, laceration and haematoma involving the skin, cramping, muscle strain, tendonitis and tendon rupture involving soft tissues, and joint sprain, dislocation and cartilage injury involving joints were used in Burgess et al. [13]. In comparison, injury type categories were limited to laceration and contusion involving the skin, sprains involving soft tissue, and dislocations involving joints in Hay et al. [19]. Dimmick et al. [6] was the only included study that did not provide data for injury type. In this review, “skin” injuries are inclusive of abrasions, lacerations, burns, haematoma and contusions, “soft tissue” injuries are inclusive of muscular strain, muscle cramping, ligament sprain/rupture, tendon sprain/rupture and tendonitis, and “joint” injuries are inclusive of dislocation, subluxation, cartilage disruption, meniscus tear, bursitis and vertebrae/facet injuries. These broad injury type groupings were based upon previous epidemiological study designs [10,18].

The large variation in skin injury frequencies among the studies may be attributed to geographic location and use of equipment, such as the protective effect of wetsuits in guarding the surfer against skin sun exposure and abrasion or laceration injury [9]. De Moraes et al. [17] conducted their survey on surfers from 3 specific cities along the Paraná seacoast compared to Furness et al. [18] recruiting participants across Australia for their survey. De Moraes et al. [17] were one of the two studies to include burns to their injury type categories which account for 23% of injuries occurring to the skin, contusions (29%) and lacerations (23%) accounting for the remaining 52%. Use of equipment is one factor none of the included studies recorded information on; the use of wetsuits in colder climate surfing may provide a protective factor against lacerations and skin injuries to surfers. 

Comparably, Donosa and Cabral [28] found lacerations and contusions to be the most common injury type, often affecting the head and lower limbs (overall injury percentages were not provided in this study). Nathanson [29] also compared outpatient to hospital-based studies; he concluded lacerations were most common in outpatient settings (35%–46% of all injuries), and fractures were most common in hospital settings with the highest percentage occurring at the face (30%). Overall percentage of fractures among all injuries was not provided [29].

### 4.3. Location of Injury

There is again variability in reporting of body region within the literature, some studies breaking body region into 18 different segments to choose from as in Burgess et al. [13] while other studies only provided five regions to choose from as in Hohn et al. [14] whose regions were limited to knee, ankle, shoulder, hip, back and other. For simplicity, this review categorized injury location into six body regions: face/head/neck, lower limb (inclusive of hip/groin to toes), upper limb/arm (inclusive of shoulder to fingers), torso (inclusive of pelvis, thorax, ribs, abdomen and chest), spine/back (inclusive of cervical, thoracic, lumbar spine and back), and other (inclusive of other and unspecified regions). Categorization of body regions was based off previous epidemiological studies designs [6,17]. All included studies provided a form of reported injury location by body region. Hohn et al. [14] provided no category for face, head or neck.

The higher proportion of face, head and neck injuries presenting to HCF could be explained by perceived severity of injury to the surfer. Face, head and neck injuries can include concussions, blunt trauma causing lacerations or contusions to the head/face which may warrant further medical investigation and contain a higher degree of perceived threat to the individual. Injury location by body region for both combined, and survey and HCF data are presented in Figure 6.

Donosa and Cabral [28] also found head and lower limbs were the most common body regions injured, within which face, neck and feet were most commonly affected. Nathanson [29] similarly found head and lower extremity were most common body regions injured. When comparing hospital-based studies to outpatient settings, facial fractures and head injuries more common in hospital presentations [29].

### 4.4. Mechanism of Injury

Injury mechanism data was limited in health care facility study data. Only two studies provided data on injury mechanism and only within the categories of struck by own board, struck by other person’s board, striking seafloor, wave turbulence and manoeuvre [6,12]. Hay et al. [19], Hohn et al. [14], Inada et al. [15], and Klick et al. [20] did not provide any information on injury mechanism in their data.

Previous reviews such as Nathanson [29] were also limited in studies which provided injury mechanism information. Donosa and Cabral [28] identified contact with the surfer’s own board as the most common mechanism of injury among their reviewed studies. Among the studies which recorded injury mechanism, all reported the majority of acute injuries were caused by collisions between the surfer and surfboard [29]. Whether this was from the surfer’s own board or another surfer’s board was not specified. The high rate of injuries caused by surfers being struck by their boards could be due to a number of different factors. Surfboards becoming projectiles during wipe outs could be the main cause of injury however, wipe outs can also be attributed to various causes. The experience level of the surfer, adherence to surf etiquette, and crowding can all contribute to causing wipe outs and increase likelihood of surfers being hit by their boards. The introduction of surfboard leashes was thought to reduce surfers being injured from other people’s boards however, also created another injury risk to surfers due to the elastic recoil of leashes during whip outs [6]. Other influencing factors could also be surfers falling onto their boards or wave force turning surfers’ own boards into projectiles during wipe outs. Environmental factors such as weather, wave size, and type of break should also be considered in influencing injury mechanism.

### 4.5. Limitations and Future Research

This review was limited by the availability of prospective surfing injury research and findings should be viewed with caution. The comparison of prospective data to retrospective data was limited by only two studies providing prospective data. The majority of studies using online surveys or questionnaires could also be considered a limitation due to the self-reported nature of online surveys. There is also increased risk of a higher number of injured surfers replying to online surveys than un-injured surfers, creating an inaccurate incidence proportion within the sport. The lack of an accurate diagnostic component by a health professional is also important to consider in online survey data. 

Further research is needed to examine surfing injuries prospectively to eliminate recall bias and better estimate the population at risk. Using data collected from medical records, ED, hospital or other health care facility data would increase the accuracy in diagnostics of injuries being reported; the creation of a global registry of injuries sustained recorded by coaches and clinicians working within the sport of surfing would also increase diagnostic accuracy and injury representation. Another limitation of this study and the included literature is the lack of information specific to geographic location of injury (i.e., reef or beach break, type of waves, etc.) was not commonly reported and therefore not analysed within this review. This could provide insight into the relationship between geographical environment and surfing injuries, beneficial to injury prevention strategies.

Based on current surfing injury trends, further research is also needed examining injury prevention strategies through use of equipment such as wetsuits, helmets and faceguards or foam-top boards, and strength and conditioning programs targeted to at risk body regions such as ankles and knees aerialists. Studies have discussed the protective effect of wetsuits and surfboard nose protectors; however, few studies have explored their use (along with other protective equipment such as helmets and faceguards) in injury prevention through prospective studies [9]. There is a need for prospective research exploring both protective equipment in injury prevention but also the effectiveness of strength and conditioning programs in the prevention of common injuries, identified through epidemiological research and systematic reviews. Injury prevention programs such as FIFA 11 in soccer have shown a decline in injury [16]. Proven strategies are yet to be established in surfing however, the identification of injury rates/proportion and location/mechanisms are now established in surfing.

## 5. Conclusions

Injuries to the skin were identified as the most common type of injury and being struck by your own board was the most common mechanism of injury both overall and in medical record data and survey data respectively. Injuries to the face, head and neck were the most common body region affected overall. Lower limb injuries followed closely as second most common injury location due to lower limb injuries being most common among survey data, representing a larger data set than health care facility data alone. Head, face and neck injuries being most common among health care facility data. This may be attributed to increased perception of injury severity with head, face and neck injuries leading to surfers seeking advice from health professionals to manage these injuries versus more self-managed lower limb injuries. These findings provide a profile of surfing injury epidemiology and highlight differences between self-reported injuries in survey data and injury data collected by health professionals through medical records. With this information, further research can be done examining the effectiveness of equipment and strength and conditioning programs in injury prevention.

## Figures and Tables

**Figure 1 sports-08-00025-f001:**
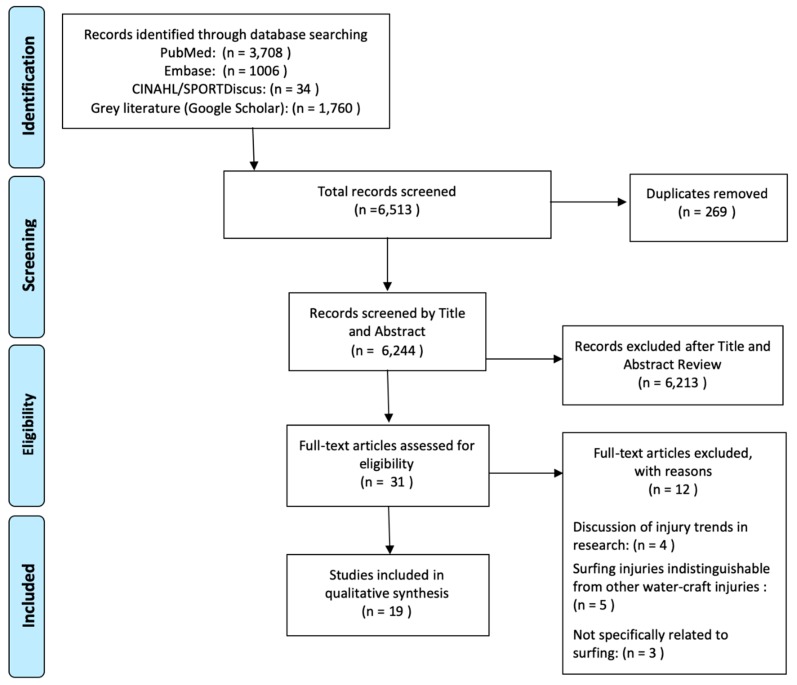
PRISMA flow diagram showing literature search, screening and eligible studies.

**Figure 2 sports-08-00025-f002:**
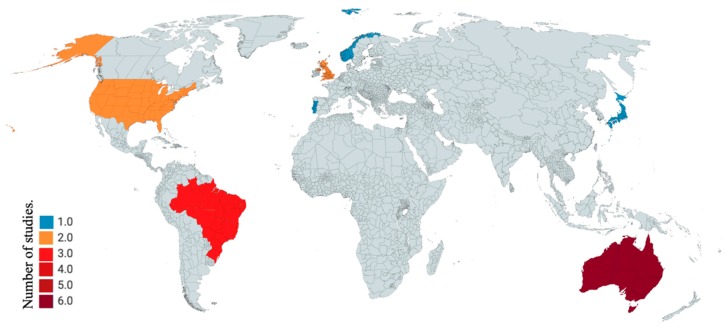
Locations of acute surfing injury data collection by country.

**Figure 3 sports-08-00025-f003:**
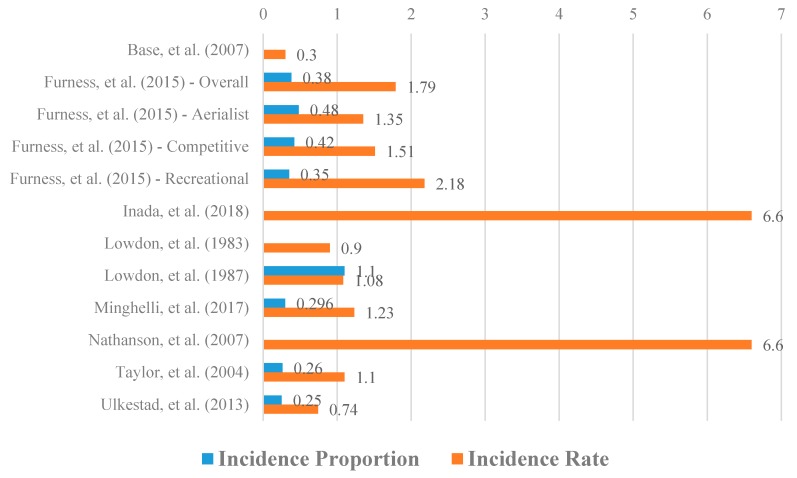
Incidence rate and proportion by article.

**Figure 4 sports-08-00025-f004:**
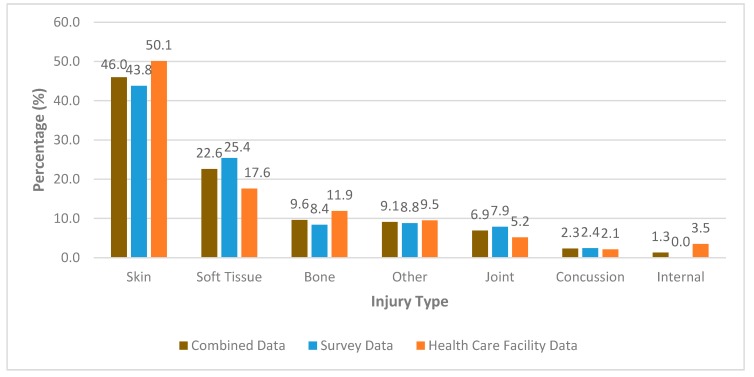
The percentage distribution of reported injury types.

**Figure 5 sports-08-00025-f005:**
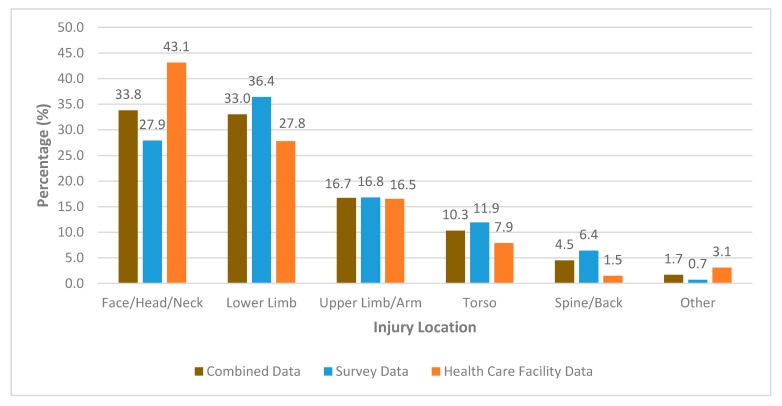
The percentage distribution of reported body regions injured.

**Figure 6 sports-08-00025-f006:**
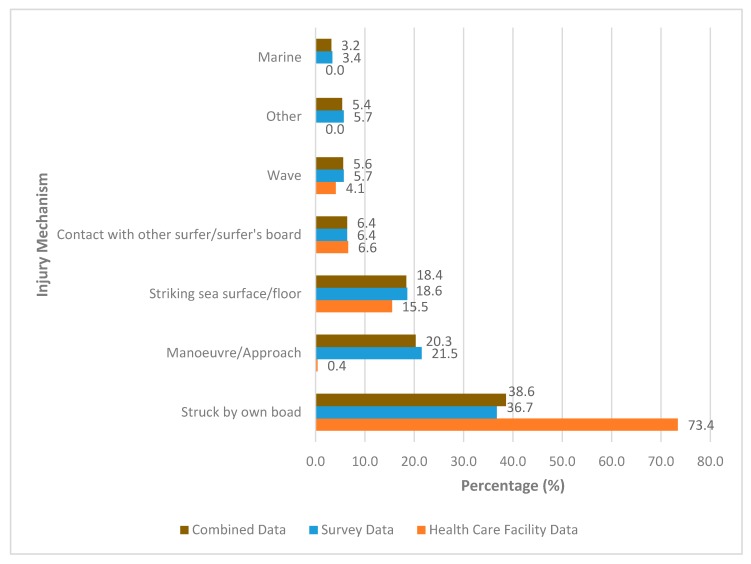
The relative percentage distribution of injury mechanism reported by survey data and emergency department/medical record data comparatively. NOTE: Mechanism data for ED studies only available for Allen et al. (1977) and Dimmick et al. (2018) with limited categories.

**Table 1 sports-08-00025-t001:** Inclusion and exclusion criteria and examples of excluded studies.

**Inclusion Criteria**	**Example/s**
Peer-reviewed journal articles	Retrieved from a scholarly journal
Surf board riding injury incidence	Studies documenting injury epidemiology: specific to injury incidence, location, types and mechanisms as a direct result of surfing
Studies documenting acute injury epidemiology	Injuries definable by a sudden onset of sharp pain or sudden impact
Musculoskeletal injury	Laceration, contusion, fracture, sprain/strain, dislocation
**Exclusion Criteria**	**Example/s**
Full text not available	Abstract
Full text not available in English	French, Spanish, German
Injuries that were not initially sustained whilst-surf board riding	Injuries initially sustained during any activity other than surfing
Non-musculoskeletal related injury or illness	Submersion, ocular trauma, exostosis etc.
Data from surfing injuries cannot be interpreted independently from other surf-sport injury data	Injury data from surfing and body surfing grouped and inseparable
Case series	A study documenting the incidence of surfing related head injuries only

**Table 2 sports-08-00025-t002:** Study design, level of evidence, Modified AXIS scores and allocated quality rating for included articles.

Author (Year)	Title	Study Design (Data Source)	Level of Evidence *	Modified AXIS	Percentage	Quality Rating
				Final	%	
Allen et al. (1977)	Surfing injuries at Waikiki	Cross-sectional Retrospective cohort (medical records)	III-2	13/24	54.2%	Fair
Base et al. (2007)	Injuries among professional surfers	Cross-sectional Retrospective cohort (questionnaire)	III-2	18/27	66.7%	Fair
Bazanella et al. (2017)	Influence of practice time on surfing injuries	Cross-sectional Retrospective cohort (questionnaire)	III-2	17/27	63.0%	Fair
Burgess et al. (2018)	An Australian survey on health and injuries in adult competitive surfing	Cross-sectional Retrospective cohort (online survey)	III-2	20/27	74.1%	Good
De Moraes et al. (2007)	Analysis of injuries’ prevalence in surfers from Parana seacoast	Cross-sectional Retrospective cohort (questionnaire)	III-2	18/27	66.7%	Fair
Dimmick et al. (2018)	Prospective analysis of surfing and bodyboard injuries	Cross-sectional Prospective cohort (ED questionnaire)	II	16/27	59.3%	Fair
Furness et al. (2015)	Acute injuries in recreational and competitive surfers: incidence, severity, location, type, and mechanism	Cross-sectional Retrospective cohort (online survey)	III-2	23/27	85.2%	Good
Hay et al. (2009)	Recreational surfing injuries in Cornwall, United Kingdom	Cross-sectional Retrospective cohort (ED questionnaire)	III-2	17/27	63.0%	Fair
Hohn et al. (2018)	Orthopedic Injuries in Professional Surfers: A Retrospective Study at a Single Orthopedic Center	Cross-sectional Retrospective cohort (Medical records – WSL)	III-2	20/24	83.3%	Good
Inada et al. (2018)	Acute injuries and chronic disorders in competitive surfing: From the survey of professional surfers in Japan	Cross-sectional Retrospective cohort (Medical records)	III-2	15/24	62.5%	Fair
Klick et al. (2016)	Surfing USA: an epidemiological study of surfing injuries presenting to US Eds 2002 to 2013	Cross-sectional Retrospective cohort (Medical records–NEISS)	III-2	16/24	66.7%	Fair
Lowdon et al. (1983)	Surfboard-riding injuries	Cross-sectional Retrospective cohort (Reply-paid questionnaire)	III-2	17/27	63.0%	Fair
Lowdon et al. (1987)	Injuries to international competitive surfboard riders	Cross-sectional Retrospective cohort(interviewed questionnaire)	III-2	18/27	66.7%	Fair
Minghelli et al. (2017)	Injuries in recreational and competitive surfers–a nationwide study in Portugal	Cross-sectional Retrospective cohort (interview questionnaire)	III-2	22/27	81.5%	Good
Nathanson et al. (2002)	Surfing injuries	Cross-sectional Retrospective cohort (online survey)	III-2	23/24	95.8%	Good
Nathanson et al. (2007)	Competitive surfing injuries: a prospective study of surfing-related injuries among contest surfers	Cross-sectional Prospective cohort (medical records)	II	19/27	70.4%	Good
Taylor et al. (2004)	Acute injury and chronic disability resulting from surfboard riding	Cross-sectional Retrospective cohort (interview questionnaire)	III-2	24/27	88.9%	Good
Ulkestad et al. (2016)	Surfing injuries in Norwegian arctic waters	Cross-sectional Retrospective cohort (online survey)	III-2	20/27	74.1%	Good
Woodacre et al. (2015)	Aetiology of injuries and the need for protective equipment for surfers in the UK	Cross-sectional Retrospective cohort (online survey)	III-2	20/27	74.1%	Good
				k = 1.000	Mean = 75.1% (SD ± 11%)	

* National Health and Medical Research Council (NHMRC) levels of evidence (Rew, 2011).

**Table 3 sports-08-00025-t003:** Method and setting of data collection, and population demographics for included studies.

Author (Year)	Data Collection Method	Data Collection Setting	Population Demographics			
			Number of Participants (N=)	$\mathrm{Mean}\;\mathrm{Age}\;(\bar{x}\;=)$	Sex (♂/♀)	Competitive Level
Allen et al. (1977)	Medical records	Waikiki Kaiser Foundation Hospital (1969–1975)	24	20 years	33/2	Recreational surfers
Base et al. (2007)	Researcher administered questionnaire	One phase of the Brazilian Professional Surfing Championship (25–26 June 2005)	32	26.5 ± 5.11 years	32/0	Professional surfers
Bazanella et al. (2017)	Researcher administered questionnaire	Subjects from Paraná coast.	66	26.16 ± 0.72 years	Unspecified	Recreational and professional surfers (min 6 months experience)
Burgess et al. (2018)	Online survey	Registered participants of Australian Surfing Titles 2014 in Coffs Harbour (1–18 August)	227	35.0 ± 13.2 years	77%/23%	Recreational surfers
De Moraes et al. (2007)	Paper back survey	Conducted on the beaches of the seacoast cities of Paraná	60	27 ± 6 years	60/0	Recreational (surfers with min. 2 years’ experience)
Dimmick et al. (2018)	ED – triage questionnaire (prospective)	Six hospitals in South East Queensland, Australia (over 18 months)	252	34 ± 12 years	89%/11%	Recreational surfers
Furness et al. (2015)	Online survey	Advertised to Australian surf websites and local surf clubs (25 October 2012, and 25 March 2013)	1348	35.8 ± 13.1 years	93.1%/6.9%	Recreational (min. 12 months of experience)
Hay et al. (2009)	ED – triage questionnaire	ED (September 2004 to August 2016).	212	27 years	80%/20%	Unspecified
Hohn et al. (2018)	Medical records	Data from the medical director of the WSL(1999 to 2016)	86	28.5 years	92.6%/7.4%	Professional surfers.
Inada et al. (2018)	Medical records	50 contests of Japan Pro Surfing Tour (2009 to 2016) and professional surfing outpatient clinic (2010 to 2016)	65	Unspecified	Unspecified	Professional surfers.
Klick et al. (2016)	Medical records	100 hospital EDs in USA (NEISS injury database; 1 January 2002 to 31 December 2013)	2072	27 years	81.9%/18.1%	Recreational surfers
Lowdon et al. (1983)	Reply-paid questionnaire	Members of the Victorian Branch of the Australian Surfriders Association (March 1982)	346	21.8 ± 5.7 years	Unspecified	Recreational surfers
Lowdon et al. (1987)	Questionnaire by interview	International surfing competitors	86	22.4 ± 3.7 years	89%/11%	Professional surfers
Minghelli et al. (2017)	Questionnaire by interview	Unspecified	1016	24.43 ± 11.98 years	84%/16%	Recreational and professional surfers
Nathanson et al. (2002)	Online survey	Advertised in periodicals and websites (May 1998 to August 1999)	1348	28.6 ± 10.6 years	90%/10%	Recreational and professional surfers.
Nathanson et al. (2007)	Medical records (prospective)	32 surf contests; 10 amateur and 22 pro contests worldwide (1999 to 2005)	116	23.6 ± 7 years	Unspecified	Recreational and professional surfers
Taylor et al. (2004)	Questionnaire by interview	Recruited beachside in Victoria (2003) and Victorian Emergency Minimum Database (VEMD)	Survey: 646 VEMD: 276	Survey: 28.2 ± 7.9 years VEMD: Unspecified	Survey: 90.2%/9.8% VEMD: 83.1%/16.9%	Survey: Unspecified VEMD: Unspecified
Ulkestad et al. (2016)	Online survey	Advertised on surfing websites and invitations to members from surfing-Facebook groups	974	Unspecified	71%/29%	Unspecified
Woodacre et al. (2015)	Online survey	Distributed to 50 surf clubs across the UK (May 2012 to November 2012);–	130	28 years	85/45	Recreational and professional surfers

**Table 4 sports-08-00025-t004:** Most common injury type, body region, and mechanism by article.

Author (Year)	Total Injuries (N=)	Type (% of Total Injuries)	Body Region (% of Total Injuries)	Mechanism (% of Total Injuries)
**Allen et al. (1977)**	**23**	**Laceration (26.1%)**	**Head (47.8%)**	**Struck by own board (91.3%)**
		**Fracture (26.1%)**	**-**	**-**
Base et al. (2007)	112	Cut/contusion (33.9%)	Lower limbs (57.6%)	Struck by own board (51.4%)
Bazanella et al. (2017)	178	Skin (46.6%)	Lower limbs (44.9%)	Struck by own board and/or seabed (40.4%)
Burgess et al. (2018)	291	Abrasion (16.5%)	Lower back (15.6%)	Struck by own board (21.5%)
De Moraes et al. (2007)	387	Contusion (29%)	Legs (26%)	Struck by own board (52%)
*** Dimmick et al. (2018)**	**248**	**-**	**Head (46.4%)**	**Struck by own board (71.8%)**
Furness et al. (2015)	512	Muscular (31.3%)	Shoulder (16.4%)	Striking seafloor (16.5%)
**Hay et al. (2009)**	**189**	**Laceration (38.6%)**	**Head (41.8%)**	**none given**
Hohn et al. (2018)	163	Ligament sprain (38.7%)	Knee (28%)	none given
Inada et al. (2018)	65	Ligament (35.1%)	Foot/ankle (40%)	none given
		Laceration (35.1%)	-	-
**Klick et al. (2016)**	**2072**	**Laceration (40.7%)**	**Lower limbs (25.9%)**	**none given**
Lowdon et al. (1983)	337	Laceration (41%)	Head (37%)	Struck by own board (45.4%)
Lowdon et al. (1987)	187	Laceration (45%)	Head (29%)	Struck by own board (35.8%)
Minghelli et al. (2017)	395	Laceration (23.5%)	Knee/leg (16.7%)	Struck by own board (27.2%)
* Nathanson et al. (2007)	116	Sprain/strain (39%)	Lower extremity (39%)	Struck by own board (29%)
Nathanson et al. (2002)	1237	Laceration (42%)	Head/neck (37%)	Struck by own board
		-	Lower extremity (37%)	-
**Taylor et al. (2004)**	165	Laceration (46.4%)	Foot/ankle (survey) (17.9%)	Struck by own board (46.1%)
	**267**	**Laceration (47.2%)**	**Face (ED) (26.6%)**	**-**
Ulkestad et al. (2013)	421	Lacerations/abrasions (30.4%)	Head/neck (43%)	Struck by own board (36.8%)
Woodacre et al. (2014)	335	Cuts/laceration (31%)	Head/face (24.2%)	Struck by own board (25.7%)

Bold = Hospital/ED records. Not bold = medical records or surveys. * = prospective study.

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
