# Peer review of "Epidemiology of Acute Injuries in Surfing: Type, Location, Mechanism, Severity, and Incidence: A Systematic Review"

_sports, 2020, doi:10.3390/sports8020025_

Round 1

Reviewer 1 Report

The authors have done a great job however I have seen some details that would improve the final result.

Title: Include “a systematic review”.

Line 17-18. However, “few studies have provided a synthesis of all available literature” no bold

Line 49. Include 1 reference supporting the paragraph.

Lines 62-66. Include each reference after the name of authors.

Line 75. Include each reference after the name of authors.

Line 74-77. The fact that it will not cover all the literature may not be enough argument to propose this review. Perhaps you should consider being an update or looking for another reason.

Line 85. Change can for could.

Line 88. Include systematic review.

Line 91. Authors should update data until November 2019

Figure 1. Change Rubmed by Pubmed

Figure 1. If I insert the review equation into pubmed (surf OR surfing) AND (acute OR trauma) AND (injury OR injuries) I only get 340 studies. Could the authors review all results?

Figure 1. If I insert the review equation into google scholar (surf OR surfing) AND (acute OR trauma) AND (injury OR injuries) I only get 41200 results. Could the authors review all results o modify the search equation?

Line 158. Include initial of two authors.

Table 3. It is very difficult to understand this table. The table include some unnecessary data. For example, title and aims of study. However, I have missed what were the outcomes measured by the different studies and the conclusions in a summary way with arrows or numbers. The conclusions of these tables are long and convoluted.

Table 3. Thinking coldly, perhaps this table could be deleted since the important data is in Table 4 and 5. Of this table little could be exploited.

Table 5. Eliminate title file.

Line 238. Eliminate reference [27].

Line 241. The authors should better explain the results of tables 4 and 5.

Line 269. Change study for systematic review.

Figure 3 to 9. I didn't know that a figure with results would be included in the discussion instead of in the results. Authors should consider this change and explain them in the results section.

Line 479. Why have the authors included the search equation in the appendix instead of in the material and methods section? Also, if this equation is included in the pubmed search engine, there are 5677 results. Revie and include this search equation in material and methods section.

Reviewer 2 Report

This research aim to systematize the available literature specific to acute surfing injuries and evaluate the differences in injury type, mechanism and location between survey and health care facility data.

The thematic is very pertinent and important for surfing enthusiasts, researchers and coaches, since is a large-scale review of acute surfing injury literature. However, in my opinion, in order to be published some changes are needed.

Try to establish a systematic use of the word surf and surfing since in all document the injuries are more related to surf and not surfing. Only two studies refer to bodyboard and in the rest is not clear the type of practice; In all document when refer to an author and colleges don’t put the citation in the end but after the reference - example: line 63 “Allen et al. who…” change to Allen et al. [1] who… Remove citations from the abstract line 32 and 33; In the literature research section is not mentioned all search made, since in figure 1 is mentioned Gray literature and other sources, and there is no information’s about this research and why the need for use other sources and in which conditions, including google scholar; Line 112 to 113 – maybe is better to reorganize the sentence something like: For ensuring the transparent and complete reporting of study selection, PRISMA statement for reporting systematic reviews and meta-analyses [21] were used, and an illustration of search results is described by the PRISMA flow diagram (Figure 1); In figure 1 change Rubmed for Pubmed in box 1 In the section Data Extraction or in the results please specify the articles that you calculate the IR; Please explain line 190 to 191 considering were in the document you provide analysis between short board and long board differences; Line 219 : please correct the introduction of the citation [27]; Please change the colors from those with only one study since is difficult to identify even knowing were the country is; Line 238: please correct the introduction of the citation [27]; Remove table 4 and reorganize line 248 to 251 since that table 5 have the same information and better organizer and easy to read. If you think that is important add another line with the total of injuries from all studies like in table 4; In table 5 footnote introduce the information about the abbreviation IP and IR Line 244 to 258 the is not results but discussion since you are comparing the studies; Line 287 remove “We found”; Line 31; 299 and 452 are you sure that the best word to use is “aerialist”/“aerialists”; The information in line 327 to 333 concerning the grouping of the injuries, specially the use of skin injuries, must be in the methods section; Please add a citation or more information, considering what is discussed in line 341 to 342 considering the geographic location since you have a beautiful graphic representation in figure 2 but nothing concerning the more common type of waves and/or beach beaks and the incidence of injuries explaining the differences ; Delete figure 5 and introduce the information in figure 4 to be easier to understand and compare the results; Delete figure 7 and introduce the information in figure 6 to be easier to understand and compare the results; Line 412 remove “This review found”; Delete figure 9 and introduce the information in figure 8 to be easier to understand and compare the results, including the appropriate space for a clear understanding of the number in the graphic; Line 429 to 430 is explored the surfer falling but nothing is mentioned about the possibility of an overcrowded surf spot and the surf etiquette respect considering the paddling for the outside and the experience of the surfers in this type of injuries; Line 435: please develop more the main idea, introduce a citation to reinforce the observation or reorganize the previous information to better tailor the core findings; In my opinion the first sentence in the conclusion is not important and should be removed; be present that the conclusion is normally the first thing to be screened in an article; Line 458 to 461: the sentence is too long, please reorganize the information During the article is not well established in all subtopic the discussion concerning the differences in injury type, mechanism and location between survey and health care facility data; A paragraph before conclusion is needed concerning the findings and the develop of injury prevention strategies for surfing population, more developed that the presented in the abstract.

Round 2

Reviewer 1 Report

The authors have made an effort to resolve my suggestions. However, they still justify the content of the tables. For me, while they don't redo the tables as I indicated in the first review, the article should not be accepted.

Table 3. It is very difficult to understand this table. The table include some unnecessary data. For example, title and aims of study. However, I have missed what were the outcomes measured by the different studies and the conclusions in a summary way with arrows or numbers.

The conclusions of these tables are long and convoluted.

Round 3

Reviewer 1 Report

The authors have done a great job and I think they have improved the manuscript. For my part, if the academic editor considers it the manuscript can be accepted.

Author Response

We thank Reviewer 1 very much for his/her suggestions to improve the manuscript.